# HO-1 Upregulation by Kaempferol via ROS-Dependent Nrf2-ARE Cascade Attenuates Lipopolysaccharide-Mediated Intercellular Cell Adhesion Molecule-1 Expression in Human Pulmonary Alveolar Epithelial Cells

**DOI:** 10.3390/antiox11040782

**Published:** 2022-04-14

**Authors:** Chien-Chung Yang, Li-Der Hsiao, Chen-Yu Wang, Wei-Ning Lin, Ya-Fang Shih, Yi-Wen Chen, Rou-Ling Cho, Hui-Ching Tseng, Chuen-Mao Yang

**Affiliations:** 1Department of Traditional Chinese Medicine, Chang Gung Memorial Hospital at Tao-Yuan, Kwei-San, Tao-Yuan 33302, Taiwan; r55161@cgmh.org.tw; 2School of Traditional Chinese Medicine, College of Medicine, Chang Gung University, Kwei-San, Tao-Yuan 33302, Taiwan; 3Department of Pharmacology, College of Medicine, China Medical University, Taichung 40402, Taiwan; lidesiao@livemail.tw (L.-D.H.); wang1203hower@gmail.com (C.-Y.W.); shihyafang@mail.cmu.edu.tw (Y.-F.S.); booden0727@gmail.com (Y.-W.C.); royeariel760918@gmail.com (R.-L.C.); huiching1205@yahoo.com.tw (H.-C.T.); 4Graduate Institute of Biomedical and Pharmaceutical Science, Fu Jen Catholic University, New Taipei City 242, Taiwan; 081551@mail.fju.edu.tw; 5Ph.D. Program for Biotech Pharmaceutical Industry, China Medical University, Taichung 40402, Taiwan; 6Department of Post-Baccalaureate Veterinary Medicine, College of Medical and Health Science, Asia University, Wufeng, Taichung 41354, Taiwan

**Keywords:** HO-1, kaempferol, human pulmonary alveolar epithelial cells, LPS, inflammation

## Abstract

Lung inflammation is a pivotal event in the pathogenesis of acute lung injury. Heme oxygenase-1 (HO-1) is a key antioxidant enzyme that could be induced by kaempferol (KPR) and exerts anti-inflammatory effects. However, the molecular mechanisms of KPR-mediated HO-1 expression and its effects on inflammatory responses remain unknown in human pulmonary alveolar epithelial cells (HPAEpiCs). This study aimed to verify the relationship between HO-1 expression and KPR treatment in both in vitro and in vivo models. HO-1 expression was determined by real time-PCR, Western blotting, and promoter reporter analyses. The signaling components were investigated by using pharmacological inhibitors or specific siRNAs. Chromatin immunoprecipitation (ChIP) assay was performed to investigate the interaction between nuclear factor erythroid-2-related factor (Nrf2) and antioxidant response elements (ARE) binding site of HO-1 promoter. The effect of KPR on monocytes (THP-1) binding to HPAEpiCs challenged with lipopolysaccharides (LPS) was determined by adhesion assay. We found that KPR-induced HO-1 level attenuated the LPS-induced intercellular cell adhesion protein 1 (ICAM-1) expression in HPAEpiCs. KPR-induced HO-1 mRNA and protein expression also attenuated ICAM-1 expression in mice. Tin protoporphyrin (SnPP)IX reversed the inhibitory effects of KPR in HPAEpiCs. In addition, in HPAEpiCs, KPR-induced HO-1 expression was abolished by both pretreating with the inhibitor of NADPH oxidase (NOX, apocynin (APO)), reactive oxygen species (ROS) (N-acetyl-L-cysteine (NAC)), Src (Src kinase inhibitor II (Srci II)), Pyk2 (PF431396), protein kinase C (PKC)α (Gö6976), p38 mitogen-activated protein kinase (MAPK) inhibitor (p38i) VIII, or c-Jun N-terminal kinases (JNK)1/2 (SP600125) and transfection with their respective siRNAs. The transcription of the *homx1* gene was enhanced by Nrf2 activated by JNK1/2 and p38α MAPK. The binding activity between Nrf2 and HO-1 promoter was attenuated by APO, NAC, Srci II, PF431396, or Gö6983. KPR-mediated NOX/ROS/c-Src/Pyk2/PKCα/p38α MAPK and JNK1/2 activate Nrf2 to bind with ARE on the HO-1 promoter and induce HO-1 expression, which further suppresses the LPS-mediated inflammation in HPAEpiCs. Thus, KPR exerts a potential strategy to protect against pulmonary inflammation via upregulation of the HO-1.

## 1. Introduction

Inflammatory components, such as adhesion molecules, are associated with the pathogenesis of lung inflammatory diseases mediated by immune reactions [1]. Moreover, various chronic pulmonary diseases, such as asthma and chronic obstructive pulmonary disease, have been linked with the upregulation of adhesion molecules [2,3,4], and the airway epithelia are mainly involved in the pathogenesis of these chronic pulmonary disorders [5,6]. Several pro-inflammatory mediators including lipopolysaccharides (LPS) could trigger acute lung inflammation during bacterial infections, through upregulation of adhesion molecules (e.g., intercellular cell adhesion protein 1 (ICAM-1) and vascular cell adhesion protein-1 (VCAM-1)) [1,2,7,8], leading to the recruitment of polymorphonuclear (PMN) cells to inflammatory tissues and airway fluid [9,10]. The upregulation of adhesion molecules is implicated in the pathological changes in the lung. Thus, either ICAM-1 or VCAM-1 could serve as an inflammatory marker, which was used to evaluate the anti-inflammation effects of heme oxygenase-1 (HO-1) in our studies. We previously revealed that LPS-induced ICAM-1 or VCAM-1 expression and monocytes (THP-1) adhesion might be used to evaluate the progression of lung inflammation [1]. Therefore, the search for therapeutic compounds protecting against ICAM-1-mediated inflammation could be an avenue in managing pulmonary diseases.

Overexpressing heme oxygenase (HO)-1 could be beneficial in the treatment of pulmonary diseases by inhibiting the expression of inflammatory mediators [8,11,12]. The rate-limiting enzyme of heme catabolism is HO, which converts heme into biliverdin, carbon monoxide, and Fe^2+^; thus far, HO-1, HO-2, and HO-3 have been recognized as the isoforms of HO [13,14]. HO-1 expression is usually low level in normal tissues, which is only induced by various mediators of redox stress and exerts anti-inflammatory and protective effects against oxidative stress [11]. Thus, searching for old compounds or developing new drugs which can induce the expression of HO-1 might be a useful strategy to protect against LPS-mediated inflammation. Tremendous studies have focused on the cellular effects of HO-1 protecting against cellular injury and oxidative stresses, however, the mechanisms engaged in the upregulation of HO-1 are still not completely understood [15]. 

Kaempferol (KPR) is a member of flavonoids, which have been recognized to possess anti-inflammation and antioxidation properties [16]. KPR has been shown to suppress LPS-induced eotaxin-1 protein expression and tumor necrosis factor (TNF)α-induced expression of epithelial ICAM-1, thus encumbering the eosinophil-airway epithelium interaction [17]. KPR also attenuates eosinophil infiltration and airway inflammation in mice sensitized with ovalbumin. Further, KPR has been shown to possess potent pharmacological activities which inhibit the levels of inflammatory mediators in guinea pigs sensitized by ovalbumin, implying that KPR is effective in the modulation of allergic respiratory diseases [18]. More recently, KPR has been demonstrated to alleviate oxidative stress and apoptosis through mitochondrion-dependent pathways during lung ischemia-reperfusion injury [19]. In different types of cells, several signaling pathways have been indicated to induce upregulation of HO-1 under various stimuli [20]. For example, AMP-activated protein kinase, mitogen-activated protein kinases (MAPKs), or nuclear factor erythroid 2-related factor 2 (Nrf2) are suggested in the involvement of HO-1 expression induced by KPR in BV-2 microglia and PC12 cells [21,22]. Our previous studies also found that nicotinamide-adenine dinucleotide phosphate (NADPH) oxidase (NOX)/reactive oxygen species (ROS)-dependent c-Src/PDGFRα/PI3K/Akt/Nrf2/antioxidant response elements (ARE) cascade [8] and protein kinase C (PKC)α/proline-rich tyrosine kinase (Pyk)2/p38α MAPK- or c-Jun N-terminal kinases (JNK)1/2-dependent c-Jun activation mediate the HO-1 expression induced by mevastatin in human pulmonary alveolar epithelial cells (HPAEpiCs) [12]. Other studies also indicated that either PKCα-dependent signaling pathways [23] or NOX/ROS/c-Src/Pyk2/Akt-dependent Nrf2 activation [24] mediates rosiglitazone-induced HO-1 expression in HPAEpiCs. Although, in various models, KPR has been extensively investigated for the effectiveness of anti-inflammation, the detailed mechanisms of HO-1 expression induced by KPR are not completely defined in HPAEpiCs and in vivo. 

In this report, the detailed mechanisms of intracellular signaling pathways involved in the KPR-induced HO-1 expression in HPAEpiCs were defined. Here, we elucidated the signaling pathways engaged in the HO-1 expression induced by KPR in HPAEpiCs. Further, we clarified whether KPR-induced HO-1 could suppress the expression of ICAM-1 in HPAEpiCs and mice triggered with LPS.

## 2. Materials and Methods

### 2.1. Reagents and Antibodies

FBS and DMEM/F-12 were purchased from Invitrogen (Carlsbad, CA, USA). GenMute siRNA Transfection Reagent was purchased from SignaGen Laboratories (Rockville, MD, USA). Kaempferol (KPR), SP600125, and tin protoporphyrin IX (SnPPIX) were purchased from Cayman Chemical (Ann Arbor, MI, USA). Anti-GAPDH (Cat# MCA-1D4, RRID:AB 2107599) mouse monoclonal antibody was from EnCor Biotechnology (Gainesville, FL, USA). Anti-ICAM-1 (EPR16608; rabbit monoclonal antibody, Cat# ab179707, RRID:AB 2814769), anti-Pyk2 (E354; rabbit monoclonal antibody, Cat# ab32448, RRID:AB_777568), anti-NOX2/gp91^phox^ (EPR6991; rabbit monoclonal antibody, Cat# ab129068, RRID:AB_11144496), anti-PKCα (phospho-S^657^)(EPR1901(2); rabbit monoclonal antibody, Cat# ab180848, RRID:AB_ 2783896), and anti-Nrf2 (phospho Ser^40^) (EP1809Y; rabbit monoclonal antibody, Cat# ab76026, RRID:AB_1524049) were purchased from Abcam (Cambridge, UK). Actinomycin D (Act.D), cycloheximide (CHI), Gö6976, p38 MAPK inhibitor (p38i) VIII, apocynin (APO), and anti-HO-1 pAb (rabbit polyclonal antibody, Cat# ADI-SPA-895, RRID:AB 10618757) were purchased from Enzo Life Sciences (Farmingdale, NY, USA). PF431396 and c-Srci II were from Merck (Merck Millipore, Billerica, MA, USA). 2’,7’-Bis-(2-carboxyethyl)-5-(and-6)-carboxyfluorescein, acetoxymethyl ester (BCECF-AM), dihydroethidium (DHE), and 2’,7’-dichlorodihydrofluorescein diacetate (H2DCFDA) were purchased from Molecular Probes (Eugene, OR, USA). Anti-p47^phox^ (phospho-Ser^370^) (rabbit polyclonal antibody, Cat# A1171, RRID:AB_10696129) antibody was from Assay Biotech (Sunnyvale, CA, USA). Anti-c-Src (SRC 2) (rabbit polyclonal antibody, Cat# sc-18, RRID:AB_631324), anti-PKCα (C20) (rabbit polyclonal antibody, Cat# sc-208, RRID:AB_2168668), anti-ICAM-1 (H-108) (rabbit polyclonal antibody, Cat# sc-137020, RRID:AB_647486), anti-Nrf2 (C-20) (rabbit polyclonal antibody, Cat# sc-722, RRID:AB_2108502), and anti-JNK1/2 (E5) (mouse monoclonal antibody, Cat# sc-7891, RRID:AB_2140720) antibodies were purchased from Santa Cruz Biotechnology (Santa Cruz, CA, USA). Anti-phospho-c-Src (Tyr^416^) (rabbit polyclonal antibody, Cat# 2101, RRID:AB_331697), anti-p47^phox^ (D21F6) (rabbit monoclonal antibody, Cat# 4301S, RRID:AB_2150286), anti-phospho-Pyk2 (Tyr^402^) (rabbit polyclonal antibody, Cat# 3291, RRID:AB_2300530), anti-p38 MAPK (D13E1) XP (rabbit monoclonal antibody, Cat# 8690, RRID:AB_10999090), anti-phospho-SAPK/JNK(Thr^183^/Tyr^185^) (mouse monoclonal antibody, Cat# 9255, RRID:AB_2307321), and anti-phospho-p38 MAPK (Thr^180^/Tyr^182^) (rabbit monoclonal antibody, Cat# 9211, RRID:AB_331641) were purchased from Cell Signaling Technology (Danvers, MA, USA). SDS-PAGE supplies were from MDBio Inc (Taipei, Taiwan). LPS (from *E. coli* serotype 0111:B4), TRIzol, 2,3-bis-(2-methoxy-4-nitro-5-sulfophenyl)-2H-tetrazolium-5-carboxanilide (XTT) assay kit, N-acetyl-L-cysteine (NAC), and other chemicals were from Sigma-Aldrich (St. Louis, MO, USA). 

### 2.2. Cell Culture and Treatment

HPAEpiCs (primary cells) isolated from human lung tissues were purchased from the ScienCell Research Laboratories (San Diego, CA, USA) and cultured in DMEM/F12 medium containing 10% FBS at 37 °C in a humidified 5% CO_2_. Cells from passages 4 to 7 were used to perform experiments, as previously described [24]. HPAEpiCs with 90% confluence were starved for 24 h by incubation in serum-free DMEM/F-12 and then challenged with 20 μg/mL LPS for the indicated time intervals after pretreated with 10 μM of KPR for 1 h.

### 2.3. Animal Care and Experimental Procedures

Animal studies are reported in compliance with the ARRIVE guidelines [25]. Male Institute of Cancer Research (ICR) mice (6–8 weeks old, weighing 18–20 g) were purchased from the National Laboratory Animal Centre (Taipei, Taiwan) and handled according to the guidelines of the Animal Care Committee of Chang Gung University (Approval Document No. CGU 16-046) and National Institutes of Health (NIH) Guides for the Care and Use of Laboratory Animals (NIH Publication No. 85-23, revised 1996). All efforts were made to reduce the number of animals used and to minimize the suffering of animals. Mice were fed with food and water ad libitum under standardized conditions (12 h light/dark cycle, 21–24 °C, humidity of 50–60%) in individually ventilated cages in a standard animal facility. Mice were randomly allocated into three groups with five mice in each group/cage: sham (0.1 mL of DMSO-PBS (1:100) with 0.1% (*w*/*v*) BSA treated mice), LPS (LPS-treated mice), and KPR + LPS (KPR plus LPS mice). ICR mice were anesthetized and individually placed on a frame in a near-vertical position and the tongues were withdrawn with lined forceps. LPS (3 mg/kg body weight) was injected into the throat and then aspirated into the lungs. Control mice were administrated with sterile 0.1% BSA in DMSO-PBS. The mice regained consciousness after 15 min. Mice were administered a dosage of KPR (0.1 mg/kg body weight) for 24 h before the LPS challenge and then killed by pentothal (i.p.; 100 mg/kg) 24 h after challenge with LPS. Lung tissues were extracted for detection of protein (right superior lobe + post-caval lobe) using a lysis buffer with pH 7.4 containing 50 mM Tris-HCl, 2 mM EDTA, 1 ug/mL aprotinin, 2 mM phenylmethylsulfonyl fluoride, and 10 ug/mL leupeptin and for detection of mRNA of ICAM-1, HO-1, or β-actin (right middle lobe + right inferior lobe) using TRIzol reagent. A tracheal cannula was adopted to collect bronchoalveolar lavage (BAL) fluid using 1 mL aliquots of ice-cold PBS solution. A Z1 Coulter Counter (Beckman Coulter, Indianapolis, IN, USA) was used to determine leukocyte count, as previously described [23]. The operators and data analysis of experiments were blinded. 

### 2.4. Immunohistochemical (IHC) Staining

IHC staining was performed on the sections of the lung tissues, which were deparaffinized, rehydrated, and washed with Tween-Tris buffered saline (TTBS). Non-specific binding was blocked by preincubation with PBS containing 5 mg/mL of BSA for 1 h at room temperature. The sections were incubated with an anti-ICAM-1 or anti-HO-1 antibody (1:100 dilution) at 4 °C for 16 h and then with an anti-mouse or anti-rabbit horseradish peroxidase antibody at room temperature for 1 h. Binding antibodies were detected by incubation in 0.5 mg/mL of 3,3-diaminobenzidine/0.01% (*v*/*v*) hydrogen peroxide in 0.1 M Tris-HCl buffer, as a chromogen (Vector Lab, Burlingame, CA, USA).

### 2.5. Protein Preparation and Western Blot Analysis

Growth-arrested cells were incubated with or without 10 μM KPR at 37 °C for the indicated time intervals. Inhibitors were added 1 h before the application of KPR, as previously described [24]. In brief, the cells were washed with cold PBS, scraped, and collected with a lysis buffer (50 mM Tris-HCl, pH 7.4, 1 mM EGTA, 1 mM NaF, 150 mM NaCl, 1 mM PMSF, 5 μg/mL leupeptin, 20 μg/mL aprotinin, 1 mM Na3VO4, 1% Triton). The levels of protein concentration were determined by a BCA reagent. Each sample was adjusted to the same protein concentration by the 5× sample buffer. The same amounts of protein (30 μg) were denatured and analyzed by 10% SDS-PAGE. Then, the proteins were transferred to the nitrocellulose membranes and probed overnight with respective primary antibodies. Membranes were washed four times with TTBS for 5 min each and incubated with an anti-rabbit or anti-mouse horseradish peroxidase antibody (1:2000 dilution) for 1 h. Finally, the immunoreactive bands were detected by ECL and captured using a UVP BioSpectrum 500 Imaging System (Upland, CA, USA). The UN-SCAN-IT gel software (Orem, UT, USA) was used to quantify image densitometry analysis. All image densitometry analyses were normalized to β-actin or respective total proteins.

### 2.6. Real-Time Quantitative PCR (RT-qPCR) Analysis

TRIzol reagent was used to extract the total RNA from HPAEpiCs which was spectrophotometrically determined at 260 nm as previously described [24]. In brief, mRNA was reverse-transcribed into cDNA and analyzed by RT-qPCR using a StepOnePlusTM real-time qPCR system (ThermoScientific-Applied Biosystems, Foster City, CA, USA) and Kapa Probe Fast qPCR Kit Master Mix (2X) Universal (KK4705; KAPA Biosystems, Wilmington, MA, USA). The levels of HO-1 and ICAM-1 expression were quantified by normalization to the level of GAPDH expression. The relative gene expression was determined using ΔΔCt method, where Ct represented the threshold cycle. The primers used in the real-time PCR reaction were as follows: ICAM-1 (NM_000201.3):Forward primer (5′ → 3′): GGCCTCAGTCAGTGTGAReverse primer (5′ → 3′): AACCCCATTCAGCGTCAHO-1 (NM_002133.3):Forward primer (5′ → 3′): CTCCCAGGCTCCGCTTCTReverse primer (5′ → 3′): GCATGCCTGCATTCACATGGAPDH (NM_001357943.2):Forward primer (5′ → 3′): GCCAGCCGAGCCACATReverse primer (5′ → 3′): CTTTACCAGAGTTAAAAGCAGCCC

### 2.7. Transient Transfection with siRNAs in HPAEpiCs

Human siRNAs of scrambled, SMARTpool RNA duplexes corresponding to PKCα (SASI_Hs01_00018816), Pyk2 (SASI_Hs01_00032249), JNK1 (SASI_Hs02_00319556), and scrambled control (negative control type 1) siRNA were from Sigma-Aldrich (St. Louis, MO, USA). JNK2 (HSS108550), p38α (HSS102352, HSS102353, HSS175313), and c-Jun (HSS105641, HSS105642, HSS180003) were from Invitrogen Life Technologies (Carlsbad, CA, USA). The transfection protocol was adopted from our previous study [24]. Briefly, Opti-MEM and Genmute reagent were used to carry out transient transfection of siRNAs. The transfection complex (siRNA 100 nM, Opti-MEM 100 μL, and Genmute reagent 2.5 μL) was incubated in the cells for 5 h. The transfection medium was replaced with a DMEM/F-12 medium containing 10% FBS overnight and then changed to a serum-free medium for 24 h. The sequences of siRNAs were listed below:Scrambled: 5′-UUCUCCGAACGUGUCACGU-3′,PKCα (NM_002737.3): 5′-AUAAGGAUCUGAAAGCCCGUUUGGA-3′,Pyk2 (NM_173174.3): 5′-CUGAUGACCUGGUGUACCU-3′,JNK1 (NM_001278547.2): 5′-GCAGAAGCAAGCGUGACAACA-3′,JNK2 (NM_001135044.2): 5′-AAUUGGUUUCAGCUGGUAACGU-3′,p38α (NM_139014.3): (1) 5′-AUGAAUGAUGGACUGAAAUGGUCUG-3′,(2) 5′-AAACAAUGUUCUUCCAGUCAACAGC-3′,(3) 5′-UUAGGUCCCUGUGAAUUAUGUCAGC-3′,c-Jun (NM_002228.4):5′-AAGUUGCUGAGGUUUGCGUAGACCG-3′,5′-AACUGCUGCGUUAGCAUGAGUUGGC-3′,5′-AUAGAAGGUCGUUUCCAUCUUUGCA-3′.

### 2.8. Chromatin Immunoprecipitation (ChIP) Assay

ChIP assay was performed to detect the interaction between human HO-1 promoter and transcription factors, as previously described [24]. A ChIP assay kit (Upstate, Lake Placid, New York, NY, USA) was used to prepare soluble chromatin according to the instructions of the manufacturer, and then soluble chromatin was immunoprecipitated without (control) or with an anti-Nrf2 antibody and normal goat IgG. PCR products for all SYBR Green primer pairs were verified to produce single products by high-resolution melt curve for avoiding the possibility of amplification artifacts. The comparative Ct method (^ΔΔ^Ct) was used to calculate the relative DNA levels. The DNA was extracted and resuspended in H_2_O and subjected to PCR amplification with the ARE primers (NC_000022.11): forward: 5′-AGAGCCTGGGGTTGCTAAGT-3′, and reverse: 5′-GGCC GGTCACATTTATGCTC-3′.

### 2.9. Transfection and Promoter Activity Assay 

ARE-luc reporter construct plasmids were transiently transfected at a concentration of 0.8 μg/mL, and the control pGal encoding for β-galactosidase presented at 0.2 μg/mL to normalize the transfection efficiency, as previously described [24]. ARE-luc luciferase activities were determined by using a luciferase assay system (Abcam, Cambridge, UK) according to the manufacturer’s instructions. Detected firefly luciferase activities were standardized with β-galactosidase activity.

### 2.10. NADPH Oxidase Activity Assay

After exposure to 10 μM KPR for the indicated time intervals, the cells were gently scraped and centrifuged at 400× *g* for 10 min at 4 °C. The cell pellet was resuspended with 35 μL of ice-cold PBS and kept on ice. The levels of NOX activity were determined as previously described [24]. To a final 200 μL volume of pre-warmed (37 °C) PBS containing either NADPH (1 μM) or lucigenin (20 μM), 5 μL of cell suspension (2 × 10^4^ cells) was added to initiate the reaction followed by immediate measurement of chemiluminescence in a luminometer (SynergyH1 Hybird Reader, BioTek, Winooski, VT, USA).

### 2.11. Measurement of Intracellular ROS Accumulation

Growth-arrested cells were treated with KPR for the indicated time intervals. When inhibitors were used, they were added 1 h before the application of KPR. The accumulation of ROS was determined as previously described [24]. After washing twice with warm PBS, the cells were stained with 10 μM H_2_DCFDA for 30 min. After staining, the cells were replaced with DMEM/F-12 containing 10% (*v*/*v*) FBS for 30 min. The fluorescence for DCF staining was detected at 495/529 nm using a fluorescence microplate reader (SynergyH1 Hybird Reader, BioTek) and FACSCalibur equipped with CellQuest software (BD Biosciences, San Jose, CA, USA). 

### 2.12. Adhesion Assay

HPAEpiCs were plated on six-well culture plates with coverslips and pretreated with or without SnPP IX, and then incubated with or without KPR for 1 h before challenge with LPS for 16 h at 37 °C in a humidified 5% CO_2_ atmosphere. THP-1 cells (human monocytic cell line, BCRC Cat# 60430, RRID: CVCL_0006) were purchased from Bioresource Collection and Research Center (Hsinchu, Taiwan) and maintained in suspension in DMEM/F-12 containing 10% (*v*/*v*) FBS medium. The number of THP-1 cells attached to HPAEpiCs was determined as previously described [24]. THP-1 cells were washed and resuspended in warm PBS, and then incubated with (BCECF-AM, 10 μM) for 1 h at 37 °C. After labeling, cells were washed thrice and resuspended in warm PBS and kept in the dark at room temperature. Then the labeled THP-1 cells were co-cultured with HPAEpiCs for 1 h in a CO_2_ incubator. These cells were gently washed with warm PBS to remove non-adherent cells. The numbers of adherent THP-1 cells were determined by counting four fields per 200× high-power field well using a fluorescence microscope with an excitation wavelength at 490 nm and an emission at 535 nm (Axiovert 200M, Zeiss, Jena, Germany). Experiments were performed in triplicate and repeated five times (*n* = 5).

### 2.13. Cell Viability Assay

According to the manufacturer’s instructions (https://www.sigmaaldrich.com/technical-documents/protocols/biology/roche/cell-proliferation-kit-xtt-assay.html, accessed on 1 April 2022), an XTT assay kit was used for cell viability and proliferation analysis.

### 2.14. Immunofluorescent Staining 

Growth-arrested cells were pretreated without or with APO, NAC, Srci II, PF431396, Gő6976, SP600125, or p38i VIII for 1 h and then incubated with KPR (10 μM) for 30 min. These cells were fixed, permeabilized, and stained using anti-p-Nrf2 antibodies (1:200 dilutions) and 4′,6-diamidino-2-phenylindole (DAPI) after washing with ice-cold PBS, and finally mounted. The images of p-Nrf2 and nucleus were detected with a fluorescence microscope (Axiovert 200 M, Zeiss, Jena, Germany).

### 2.15. Data and Statistical Analysis

The data and statistical analysis obeyed the suggestions on experimental design and analysis and reporting in pharmacology [26]. GraphPad Prism Program 6.0 software (GraphPad, San Diego, CA, USA) was used to perform statistical analysis. We used one-way ANOVA followed by Dunnett’s post hoc test when comparing more than two groups of data as previously described [24]. *p*-Values of 0.01 were statistically significant. Post hoc tests were analyzed only if F achieved *p* < 0.01 and there was no variance inhomogeneity. Error bars were omitted when they fell within the dimensions of the symbols. All the data were expressed as mean ± SEM, in five individual experiments (*n* = 5).

## 3. Results

### 3.1. KPR Upregulates HO-1 and Reduces ICAM-1 Expression Induced by LPS

To examine the role of KPR-induced HO-1 expression in LPS-induced ICAM-1 expression in HPAEpiCs, cells were incubated with 10 μM of KPR for 1 h and challenged with 20 μg/mL LPS for the various time courses. KPR stimulated protein expression of HO-1 through the time intervals and LPS time-dependently induced ICAM-1 expression with a peak increase between 6 and 24 h which was blocked by KPR treatment determined by Western blot (Figure 1A). To further examine the effects of KPR on gene transcription of itself *homx1* and LPS-induced *ICAM-1*, the levels of these mRNA expressions were analyzed by real-time PCR. HPAEpiCs were incubated with LPS for 4 h after pretreatment with 10 μM KPR for 1 h. KPR simultaneously enhanced mRNA expression of HO-1 and reduced ICAM-1 mRNA expression stimulated by LPS (Figure 1B). Additionally, by HO-1 siRNA the effect of KPR was certified. Pretreatment with KPR as indicated time intervals attenuated ICAM-1 protein level, which was reversed by HO-1 siRNA transfection under KPR pretreatment for 1 and 8 h (Figure 1C). Besides, pretreatment with 10 μM KPR for 1 or 8 h attenuated the LPS-induced cell adhesion within 16 h (Figure 1D). Moreover, we revealed that the inhibitory effect of monocyte adhesion was caused by HO-1 upregulation which was reversed by SnPPⅨ (1 μM) pretreatment. Finally, we demonstrated that adhesion of THP-1 to HPAEpiCs challenged with LPS was enhanced, which was inhibited by an ICAM-1-neutralizing antibody (Figure 1D). These results suggested that KPR-mediated HO-1 upregulation could attenuate the LPS-stimulated ICAM-1 increase and monocyte adhesion in HPAEpiCs.

### 3.2. KPR Upregulates HO-1 to Reduce LPS-Induced ICAM-1 Expression In Vivo

Further, we confirmed the KPR effect on HPAEpiCs through an animal study. As shown in Figure 2A, the IHC staining demonstrated that HO-1 expression upregulated by KPR intervention reduced the levels of ICAM-1 expression induced by LPS. Besides, in in vivo study, ICAM-1 protein expression in the lung from KPR + LPS groups was inhibited by HO-1 expression induced by KPR (Figure 2B). These findings were supported by RT-PCR showing a significant enhancement of ICAM-1 mRNA expression by LPS, which was inhibited by KPR mediated through induction of HO-1 mRNA expression in the lung tissues (Figure 2C). Our data also demonstrated that the increased number of leukocytes in the BAL fluid in mice stimulated with LPS were attenuated by KPR pretreatment (Figure 2D). These results suggested that KPR protects against the LPS-stimulated pulmonary inflammation mediated through the upregulation of HO-1. 

### 3.3. KPR Activates Transcriptional and Translational Processes to Induce HO-1 Protein and mRNA Expression 

To investigate whether KPR induced the expression of HO-1, HPAEpiCs were stimulated with KPR for the various time courses. As shown in Figure 3A, KPR induced a time- and concentration-dependent HO-1 protein expression determined by Western blot. Compared with control, there was a significant rise (2.5–7.0-fold) in HO-1 protein expression within 8–24 h, when the cells were exposed to 10 μM. Additionally, we found that KPR had no significant effect on cell viability under 30 μM within 24 h, analyzed by an XTT assay kit (Figure 3B). Hence, for performing the following experiments, 10 μM KPR was used throughout this study.

To examine the effects of *homx1* gene transcription and translation by KPR, the promotor activity and the levels of HO-1 mRNA expression were determined by luciferase promotor report assay and real-time PCR, respectively. HPAEpiCs were treated with KPR (10 μM) for the indicated time courses. KPR induced time-dependent expression of HO-1 mRNA and promotor activity within 4–8 h (Figure 3C). Further, HPAEpiCs were incubated with 10 μM KPR for 6 h (mRNA expression) or 16 h (protein expression) after pretreatment with actinomycin D (Act. D) or cycloheximide (CHI) for 1 h. Either Act. D or CHI pretreatment concentration-dependently decreased the KPR-induced HO-1 protein expression (Figure 3D). Besides, KPR-induced mRNA expression of HO-1 was attenuated by Act. D but not CHI (Figure 3E). These results suggested that in HPAEpiCs, KPR-induced HO-1 protein expression is dependent on the process of *homx1* gene transcription and translation.

### 3.4. KPR Stimulates NOX2 and Enhances ROS Generation to Induce HO-1 Expression 

ROS could act as second messengers in the upregulation of HO-1 expression [27]. NAC was used to scavenge ROS to examine whether ROS generation is engaged in the KPR-induced HO-1 expression. The KPR-induced HO-1 protein, mRNA expression, and promoter activity were concentration-dependently attenuated by pretreatment with NAC (Figure 4A,B). Numerous studies have proven that ROS production is mediated through NOXs activity [28,29]. p47^phox^ is a key component in the activation of NOX. To investigate the role of p47^phox^ in the KPR-mediated responses, apocynin (APO; an inhibitor of p47^phox^) was used. Pretreatment with APO concentration-dependently inhibited the HO-1 protein, mRNA expression, and promoter activity induced by KPR (Figure 4A,B). KPR also time-dependently enhanced NADPH oxidase activity and ROS generation (Figure 4C), which was significantly attenuated by APO or NAC pretreatment (Figure 4D) determined by NADPH oxidase activity assay and a fluorescent probe H_2_DCF-DA, respectively. To determine which isoform of NOXs participated in KPR response and verify the effect of p47 in the KPR-induced HO-1 expression, as shown in Figure 4E, compared with scrambled siRNA, cells transfected by NOX2 or p47 siRNA attenuated the KPR-induced HO-1 protein level but not by NOX1, 3, or 4 siRNA (data not shown). Further, to explore whether p47 phosphorylation was necessary for HO-1 expression, 10 μM KPR was used to stimulate HPAEpiCs for the indicated time intervals. As shown in Figure 4F, KPR stimulated p47 phosphorylation in a time-dependent manner which was attenuated by p47 siRNA transfection. These results suggested that the NOX2/ROS-dependent mechanism is involved in the induction of HO-1 by KPR in HPAEpiCs. 

### 3.5. c-Src Is Needed for HO-1 Expression Induced by KPR

The c-Src activity could enhance HO-1 expression [8]. We evaluated the effect of Src inhibitor II (Srci II) on the KPR-induced HO-1 expression to determine whether c-Src was required in the KPR induced HO-1 expression in HPAEpiCs. As shown in Figure 5A,B, Srci II caused a concentration-dependent decrease in the KPR-induced HO-1 protein, mRNA expression, and promoter activity. Transfection with c-Src siRNA was used to ensure the role of c-Src in the KPR-induced HO-1 expression, as shown in Figure 5C, downregulation of c-Src protein by c-Src siRNA attenuated the KPR-induced HO-1 protein expression. Further, HPAEpiCs were treated with 10 μM KPR for the indicated time intervals to investigate whether c-Src phosphorylation was required for HO-1 expression. Transfection with c-Src, p47, or NOX2 siRNA time-dependently attenuated the c-Src phosphorylation stimulated by KPR (Figure 5D). These results suggested that activation of NOX2/ROS/c-Src regulates the KPR-induced HO-1 expression in HPAEpiCs.

### 3.6. Pyk2 Is Involved in KPR-Induced HO-1 Expression

Activation of Pyk2 could increase HO-1 expression [12]. The effect of PF431396 on the KPR-induced HO-1 expression was evaluated to determine whether Pyk2 was involved in the HO-1 expression induced by KPR in HPAEpiCs. Pretreatment with PF431396 concentration-dependently decreased HO-1 protein, mRNA expression, and promoter activity induced by KPR (Figure 6A,B). Pyk2 siRNA was used to ascertain the role of Pyk2 in the KPR-induced HO-1 expression. As shown in Figure 6C, Pyk2 protein was knocked down by transfection with Pyk2 siRNA, which attenuated the KPR-induced HO-1 protein expression in HPAEpiCs. Further, HPAEpiCs were stimulated with 10 μM KPR for the indicated time intervals to investigate whether phosphorylation of Pyk2 was necessary for HO-1 expression. As shown in Figure 6D, siRNA of c-Src or Pyk2 attenuated the time-dependent increase of Pyk2 phosphorylation stimulated by KPR. Transfection with Pyk2 siRNA had no significant effect on KPR-stimulated c-Src phosphorylation, implying that Pyk2 was a downstream component of c-Src in KPR-mediated responses. These results suggested that activation of c-Src/Pyk2 mediates the KPR-induced HO-1 expression in HPAEpiCs.

### 3.7. PKCα Is Needed for HO-1 Expression Induced by KPR

Activation of PKCα could enhance HO-1 expression [12]. Gö6976 was used to determine whether PKCα was involved in the HO-1 expression induced by KPR in HPAEpiCs. As shown in Figure 7A,B, KPR-induced HO-1 protein, mRNA expression, and promoter activity were concentration-dependently decreased by pretreatment with Gö6976. PKCα siRNA was used to ensure the role of PKCα in the KPR-induced HO-1 expression. As shown in Figure 7C, PKCα protein was knocked down by transfection with PKCα siRNA, which attenuated the HO-1 protein expression induced by KPR. Further, HPAEpiCs were treated with 10 μM KPR for the indicated time intervals to investigate the role of PKCα phosphorylation in HO-1 expression. As shown in Figure 7D, PKCα phosphorylation was time-dependently stimulated by KPR, which was attenuated by transfection with either Pyk2 or PKCα siRNA. These results suggested that activation of Pyk2/PKCα regulates the KPR-induced HO-1 expression in HPAEpiCs.

### 3.8. KPR Induces HO-1 Expression Mediated through p38 MAPK and JNK1/2

Activation of MAPKs could enhance HO-1 expression [12,23]. We evaluated the effect of p38i VIII or SP600125 on the HO-1 expression induced by KPR to determine whether p38 MAPK and JNK1/2 were involved in the KPR induced HO-1 expression in HPAEpiCs. As shown in Figure 8A,B, as well as Figure 9A,B, HO-1 protein, mRNA expression, and promoter activity induced by KPR were concentration-dependently decreased by pretreatment with either p38i VIII or SP600125. siRNA of p38, JNK1, or JNK2 was used to ensure the role of p38 MAPK or JNK1/2 in the KPR-induced HO-1 expression. As shown in Figure 8C and Figure 9C, in HPAEpiCs, p38, JNK1, or JNK2 siRNA was knocked down by transfection with individual siRNA, which attenuated the HO-1 protein expression induced by KPR. Further, HPAEpiCs were treated with 10 μM KPR for the indicated time intervals to investigate whether phosphorylation of p38 or JNK1/2 was engaged for HO-1 expression. As shown in Figure 8D and Figure 9D, p38 or JNK1/2 phosphorylation was time-dependently induced by KPR. Transfection with p38 or PKCα siRNA attenuated phosphorylation of p38 MAPK induced by KPR. Besides, transfection with JNK1, JKN2, or PKCα siRNA attenuated KPR-induced the phosphorylation of JNK1/2. These results implied that activation of PKCα/p38 MAPK or JNK1/2 regulates the HO-1 expression induced by KPR in HPAEpiCs.

### 3.9. KPR Stimulates Phosphorylation of Nrf2 Involved in HO-1 Expression

It is reported in literature that Nrf2 activity could promote the ARE-driven antioxidant enzymes expression, including HO-1, which can protect against cellular injury and toxicity [8,24]. Nrf2 siRNA was transfected to investigate whether Nrf2 regulates the KPR-induced HO-1 expression in HPAEpiCs. Nrf2 protein level was knocked down by transfection with Nrf2 siRNA, which attenuated the HO-1 protein (Figure 10A) and mRNA (Figure 10B) expression induced by KPR. To further examine whether Nrf2 and its upstream components were involved in the KPR-mediated transcriptional activity via binding with ARE in the promoter region of the *HM**OX1* gene in HPAEpiCs, ARE promoter activity and ChIP were performed. As shown in Figure 10C, KPR progressively promoted ARE promoter activity with a peak response within 6 h, which was blocked by pretreatment with APO, NAC, or Srci II. Moreover, the ChIP assay further revealed that KPR time-dependently enhanced Nrf2 bound with ARE1 on the promoter of HO-1 reaching a maximal level within 1 h, which was significantly blocked by pretreatment with APO, NAC, Srci II, PF431396, or Gö6976 (Figure 10D). 

Furthermore, we found that KPR promoted the phosphorylation of Nrf2 in a time-dependent manner, which was inhibited by Nrf2, p38 MAPK, or JNK2 siRNA transfection (Figure 10E). We also performed experiments to investigate the translocation of phospho-Nrf2 into the nucleus by immunofluorescence staining. Our results revealed that KPR stimulated Nrf2 phosphorylation and translocation into the nucleus within 1 h, which were inhibited by APO, NAC, Srci II, PF431396, Gő6976, SP600125, or p38i VIII (Figure 10F). These results indicated that KPR upregulates HO-1 expression via activation of NOX/ROS/c-Src/Pyk2/PKCα cascade leading to Nrf2 phosphorylation and binding with the HO-1 promoter of the ARE binding sites in HPAEpiCs.

## 4. Discussion

HO-1 is well-known as an inducible isoform of three subtypes characterized in humans [14]. It has a protective role in defending against inflammation-mediated and oxidation-induced cellular damage and tissue injury. Several animal studies have indicated that HO-1 may attenuate lung injury in various models [13]. However, the detailed molecular mechanisms of KPR-mediated anti-inflammatory function and HO-1 upregulation remain unknown in HPAEpiCs. In this study, using pharmacological inhibitors or transfection with siRNAs uncovered the KPR-mediated responses. The dose of KPR used in the in vivo study may be safe. Herein, our main findings established that HO-1 induced by KPR is mediated through NOX2/ROS/c-Src/Pyk2/PKCα/MAPKs pathway-dependent Nrf2 activation and protects against the lung inflammation caused by LPS-induced ICAM-1 expression (Figure 11).

LPS is known as a mediator participating in the process of immune responses and infection via Toll-like receptor 4-related signaling pathways [30]. LPS stimulates the upregulation of inflammatory mediators, such as ICAM-1, in a variety of cell types including pulmonary alveolar epithelial cells [1,7]. The upregulation of ICAM-1 plays a crucial role in the pathogenesis of different diseases [3,4,31]. Therefore, ICAM-1 represents an effective target for anti-inflammatory therapies in pulmonary inflammation. Overexpression of HO-1 induced by IL-10 can rescue the LPS-mediated septic shock in mice [32]. Park et al. (2013) also found that compounds of *Inula helenium* L. extracted by ethanol activate p38 MAPK/Nrf2/HO-1 cascade and inhibit the LPS-mediated phosphorylation of IκBα and levels of adhesion molecules (VCAM-1 and ICAM-1) induced by TNF-α in RAW264.7 cells [33]. The pleiotropic effects of HO-1 on organ protection make it an important component in the treatment of various diseases [11]. Moreover, several pharmacological compounds have been proved to be beneficial for therapeutic efficacy, at least in part, due to HO-1 induction [34]. KPR has pleiotropic effects including anti-inflammation and anti-oxidation [16]. The advantage of KPR could arise from its effects on modulating some signal components linked to inflammation, cellular apoptosis, metastasis, and angiogenesis [16]. A prospective study indicated that dietary intake of KPR has a significant 40% decrease in ovarian cancer incidence [35]. Moreover, a growing body of literature has shown that KPR in in vivo animal studies could be used as complementary medicine for the prevention and treatment of different illnesses, such as myocardial infarction [36], obesity and diabetes [37], arthritis [38], Parkinson’s disease [39], asthma [40,41], thromboembolism [42], osteoporosis [43], etc. These findings are comparable to our findings suggesting that HO-1 expression induced by KPR protects against the ICAM-1 expression induced by LPS. SnPPIX (an HO-1 enzyme inhibitor) reversed the protective effect of KPR in HPAEpiCs. The natural source, low-cost relative to synthetic drugs, and safety of KPR have been elucidated in several lines of literature [44]. In this study, KPR below 30 μM within 24 h also had no significant effect on cytotoxicity, as previously reported. Thus, KPR might be a potential therapeutic approach for inflammatory diseases of the lung.

This study found that in HPAEpiCs, KPR significantly induces the levels of HO-1 protein and mRNA expression. Our data also indicated that the HO-1 protein upregulated by KPR is mediated via a de novo HO-1 mRNA synthesis which was attenuated by Act. D but not by cycloheximide in HPAEpiCs. These results implied that the transcriptional level is primarily involved in the KPR-induced *homx1* gene expression. The dosage of KPR adopted had no significant effect on the cell viability of HPAEpiCs.

One of the major intracellular sources of ROS is the NOX family. In the normal physiological functions and the inflammatory responses, ROS dependent on their cellular concentrations could act as a messenger or mediator [45]. In both in vitro and in vivo studies, various stimuli can generate NOX-dependent ROS and further induce HO-1 expression [8,24,28]. Therefore, the upregulation of HO-1 expression by KPR, at least in part, is regulated by NOX-dependent ROS generation in HPAEpiCs. We further dissected the roles of NOX/ROS on the KPR-mediated HO-1 expression in HPAEpiCs. We used a thiol-containing compound (NAC) to scavenge ROS including hydroxyl radicals and hydrogen peroxide. In HPAEpiCs, NAC has been well known to protect cells against hydrogen peroxide-induced injury [46]. Our data also showed that KPR-induced HO-1 expression was attenuated by NAC or NADPH oxidase inhibitor (apocynin). These results strongly implied the roles of NOX/ROS in HO-1 expression induced by KPR. Our results revealed that pretreatment with APO (an inhibitor of p47^phox^) could attenuate the KPR-induced ROS generation. p47^phox^ is one of the cytosolic proteins of NOX2 containing p47^phox^, p67^phox^, p40^phox^, a GTPase Rac1 or Rac2, which are recruited into membrane sites upon activation. Further, NAC (ROS scavenger) can inhibit the KPR-mediated HO-1 expression by reducing the ROS generation. We ensured NOX/ROS involved in KPR-stimulated HO-1 by transfecting HPAEpiCs with either p47^Phox^ or NOX2 siRNA, in these cells, HO-1 protein expression induced by KPR was significantly attenuated. In contrast, recent studies showed that KPR reduced ROS levels by inhibiting NOX activity which protected against oxidative stress [47,48,49]. Our study indicated that KPR treatment activated NOX activity to increase ROS levels, in HPAEpiCs, leading to the upregulation of HO-1. The discrepancy may come from different types of cells and experimental conditions. These findings suggested that HO-1 expression may be enhanced by NOX2/ROS generation in HPAEpiCs stimulated with KPR, consistent with our previous reports [24,28]. However, we noted that NADPH oxidase activity declined, and ROS generation was still increased within 2 h. The difference between NADPH oxidase activity and ROS generation was complicated. Based on the literature, ROS production mainly occurs through mitochondrial oxidative phosphorylation and membrane-bound enzyme NADPH oxidase [50]. Other enzymes such as xanthine oxidase, cyclooxygenases, and nitric oxide synthase have also been implicated in the ROS generation induced by various ligands. In this study, NADPH oxidase-derived ROS generation was investigated, however, other ROS sources such as mitochondria are also involved in the KPR-mediated responses. The roles of other sources in the generation of ROS may be a limitation in this study and should further be investigated in the future.

In various cellular functions, the PKC family plays an important role in regulating multi-signal transductions [51]. Among this family, the most prominent modulator of HO-1 expression is PKCα, which is an upstream molecule of Nrf-2, in RAW264.7 macrophages, to protect the apoptosis triggered by LPS [52]. The present study also indicated that Gö6976 or transfection with PKCα siRNA attenuated the KPR-induced HO-1 expression. KPR-induced responses were mediated through the phosphorylation of PKCα which was inhibited by transfection with Pyk2 or c-Src siRNA, suggesting that c-Src/Pyk2 are upstream components of PKCα. However, a previous study showed that c-Src/Pyk2 are stimulated by PKC activity, as downstream components [53]. The contrast signaling cascade may result from different types of cells and stimuli. Our previous studies have demonstrated that HO-1 upregulation mediated by activation of c-Src/Pyk2 protects against lung inflammation [8,12,24]. The current study reveals that c-Src/Pyk2 plays a crucial role in the KPR-induced HO-1 expression because Src inhibitor Ⅱ or PF431396, as well as transfection with c-Src or Pyk2 siRNA, blocked HO-1 expression induced by KPR. Transfection with either c-Src or Pyk2 siRNA also attenuated the KPR-stimulated phosphorylation of Pyk2. In contrast, phosphorylation of c-Src was not inhibited by transfection with Pyk2 siRNA, suggesting that c-Src is an upstream component of Pyk2. These results proved that c-Src/Pyk2-dependent PKCα activity is involved in the mechanisms underlying KPR-induced HO-1 expression in HPAEpiCs.

Upregulation of HO-1 expression can be mediated through MAPKs upon exposure to diverse stimuli [12,23]. Our previous study has shown that upregulation of HO-1 mediated through activation of JNK1/2 and p38 MAPK can protect against the TNFα-induced pulmonary inflammation [12]. Moreover, fisetin, one of the flavonoids, induces *homx1* gene transcription in human umbilical vein endothelial cells mediated through p38 MAPK activation [54]. Sulfuretin, another flavonoid, has also been revealed to upregulate the expression of HO-1 dependent on the JNK1/2 signaling, which in turn rescues oxidative injury in human liver-derived HepG2 cells [55]. The ERK1/2, another member of MAPKs, has been clarified to participate in HO-1 induction in various types of cells by a variety of inducers [56,57,58]. These findings have uncovered that these MAPKs signalings could mediate HO-1 upregulation by some components extracted from the herbs. Our study also indicated that in HPAEpiCs, JNK1/2 and p38 MAPK were involved in the HO-1 expression induced by KPR. This hypothesis was elucidated by data showing that their pharmacological inhibitors or transfection with either p38 MAPK or JNK1/2 siRNA attenuated the induction of HO-1 by KPR and phosphorylation of Nrf2. On the contrary, a growing body of evidence has indicated that KPR has an inhibitory effect on the activity of ERK1/2 [42,59,60,61]. Indeed, our study showed that pretreatment with various concentrations of U0126 (0.1, 1, 10 μM) had no significant effect on the KPR-induced HO-1 expression in HPAEpiCs. Further, we also found that KPR did not significantly stimulate the phosphorylation of ERK1/2 during the period of observation. These results indicated that KPR-induced HO-1 expression was independent on ERK1/2 in these cells (data not shown). Therefore, our findings suggested that JNK1/2 and p38 MAPK participate in the expression of HO-1 in HPAEpiCs challenged with KPR via activating Nrf2 activity. 

The Keap1-Nrf2 pathway responds to ROS- and electrophiles-mediated endogenous and exogenous stresses [62], leading to dissociation of Nrf2 from Keap1 complex due to phosphorylation of Nrf2 or the modification of –SH group on Keap1. Nrf2 functions as a major cytoprotective regulator. The liberated phospho-Nrf2 moves into the nuclear compartment and binds with the promoter region of ARE sequences, then leading to increased Nrf2-regulated transcription of phase II antioxidant enzymes, such as HO-1 [53,63]. Although oxidative stress means the elevation of the levels of intracellular ROS that destroys the cell structure, including DNA, protein, and lipid. Nevertheless, growing evidence has indicated that a slight rise in ROS is required to maintain biological and physiological processes via activation of various cellular signalings. Our study found that in HPAEpiCs, phospho-Nrf2 was accumulated in the nucleus to participate in the HO-1 expression induced by KPR. Moreover, we further elucidated that APO, NAC, Gö6976, Srci II, or PF431396 inhibited the interaction between Nrf2 and the ARE binding site of the HO-1 promoter. Therefore, our findings suggested that NOX/ROS/c-Src/Pyk2/PKCα/MAPKs cascade activates Nrf2 which may bind to the HO-1 promoter of the ARE sequence, finally leading to the induction of HO-1 expression in HPAEpiCs.

## 5. Conclusions

These findings suggest that KPR-induced HO-1 expression might protect lung inflammation via suppressing the LPS-mediated inflammation in HPAEpiCs and in vivo. Thus, KPR exerts a potential strategy for protecting against lung inflammation. In summary, KPR activates NOX/ROS/c-Src/Pyk2/PKCα/JNK1/2- and p38α MAPK-dependent Nrf2 pathway, which further binds with ARE of HO-1 promoter. Although many mechanisms should be clarified henceforth, our report suggested that KPR could be a potential intervention for inflammatory diseases of the lung. The limitation of in vitro use of THP-1 as an adhesion assay warrants further exploration of primary monocytes in the future. In this study, the changes in the levels of ICAM-1 were used as an indicator of functional activity of HO-1 expression by kaempferol on LPS-induced pulmonary inflammation. We did not explore the detailed mechanisms by which HO-1 inhibits LPS-induced responses. It is an important issue for further study in the future. This is also a limitation of the present study.

## Figures and Tables

**Figure 1 antioxidants-11-00782-f001:**
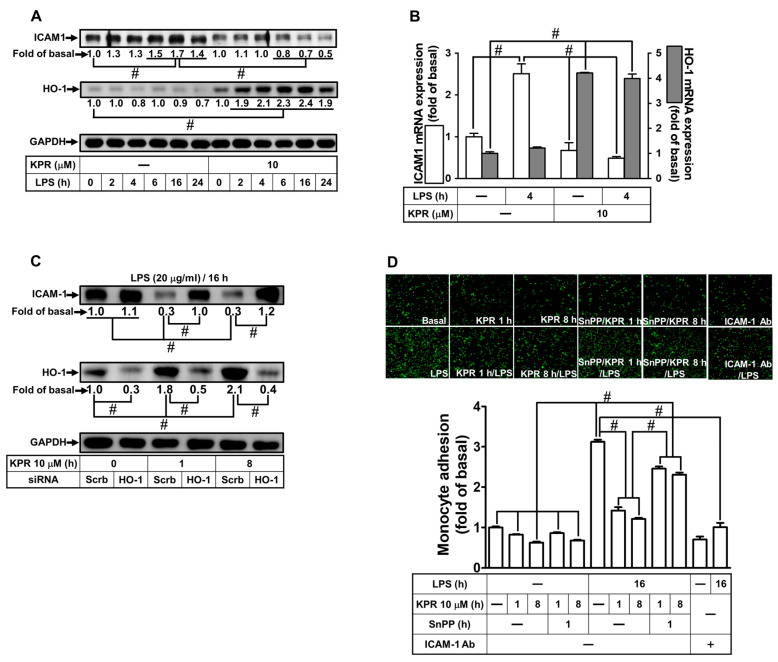
KPR inhibits ICAM-1 expression induced by LPS via HO-1 upregulation in HPAEpiCs. (**A**) HPAEpiCs were pretreated with KPR (10 μM) for 1 h, and then incubated with LPS (20 µg/mL) for the indicated time intervals. The protein levels of ICAM-1 and HO-1 were determined by Western blot using GAPDH as a loading control. (**B**) Cells were pretreated with KPR (10 μM) for 1 h, and then incubated with LPS (20 μg/mL) for 4 h. The levels of ICAM-1 and HO-1 mRNA were determined by real-time PCR. (**C**) Cells were transfected with scrambled or HO-1 siRNA, treated with KPR (10 μM) for 1 or 8 h, and then incubated with LPS (20 μg/mL) for 16 h. The levels of ICAM-1 and HO-1 protein were determined by Western blot using GAPDH as a loading control. (**D**) Cells were pretreated KPR for 1 h, then incubated by ZnPPIX for 1 h, and finally stimulated with LPS for 16 h. In addition, cells were incubated with LPS for 16 h and treated with an anti-ICAM-1 neutralizing antibody (2 μg/mL) for 1 h. The adhesion of THP-1 cells (displayed in green) was measured. Data are expressed as mean ± SEM (*n* = 5), analyzed with one way ANOVA and Dunnett’s post hoc test. # *p* < 0.01, as compared with the cells exposed to vehicle alone; or significantly different as indicated.

**Figure 2 antioxidants-11-00782-f002:**
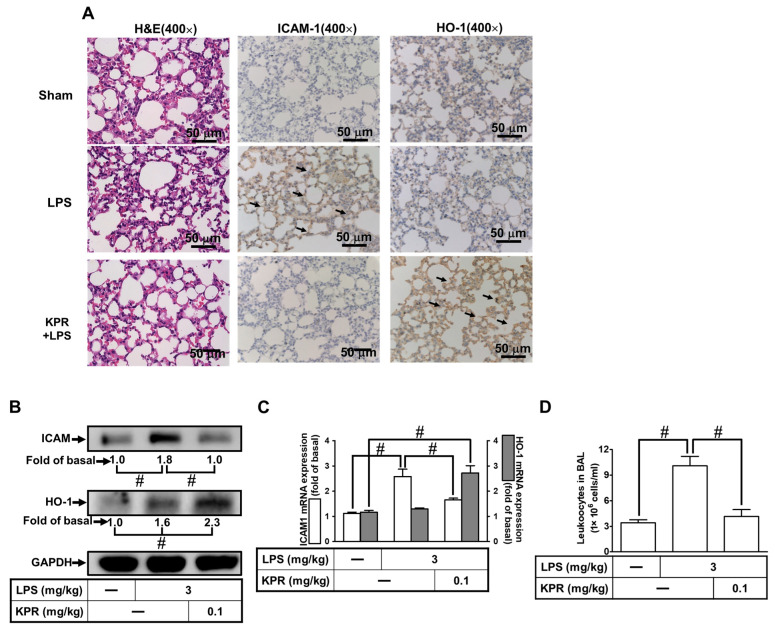
Upregulation of HO-1 by KPR attenuates LPS-induced pulmonary inflammatory responses in vivo. (**A**) Mice were intra-peritoneally pretreated with KPR (0.1 mg kg^−1^) or vehicle for 1 h, and then intratracheally administered with or without LPS (3 mg kg^−1^) for 16 h. H&E and immunohistochemical staining for ICAM-1 and HO-1 in serial sections of the lung tissues from Sham (0.1 mL of DMSO-PBS (1:100) with 0.1% (*w*/*v*) BSA treated mice), LPS (LPS-treated mice), and KPR + LPS (kaempferol plus LPS mice). The arrows indicate the ICAM-1 and HO-1 expression on pulmonary alveolar cells. All images are representative of five mice per group. (**B**,**C**) Lung tissues were homogenized to extract protein and mRNA. The levels of ICAM-1 and HO-1 protein (**B**) and mRNA (**C**) were determined by Western blot and real-time PCR. (**D**) Leukocyte count in the bronchoalveolar lavage fluid (BALF) of sham, LPS, and KPR + LPS groups. Data are expressed as mean ± SEM (*n* = 5), analyzed with one way ANOVA and Dunnett’s post hoc test. # *p* < 0.01, as compared with the cells exposed to vehicle alone; or significantly different as indicated.

**Figure 3 antioxidants-11-00782-f003:**
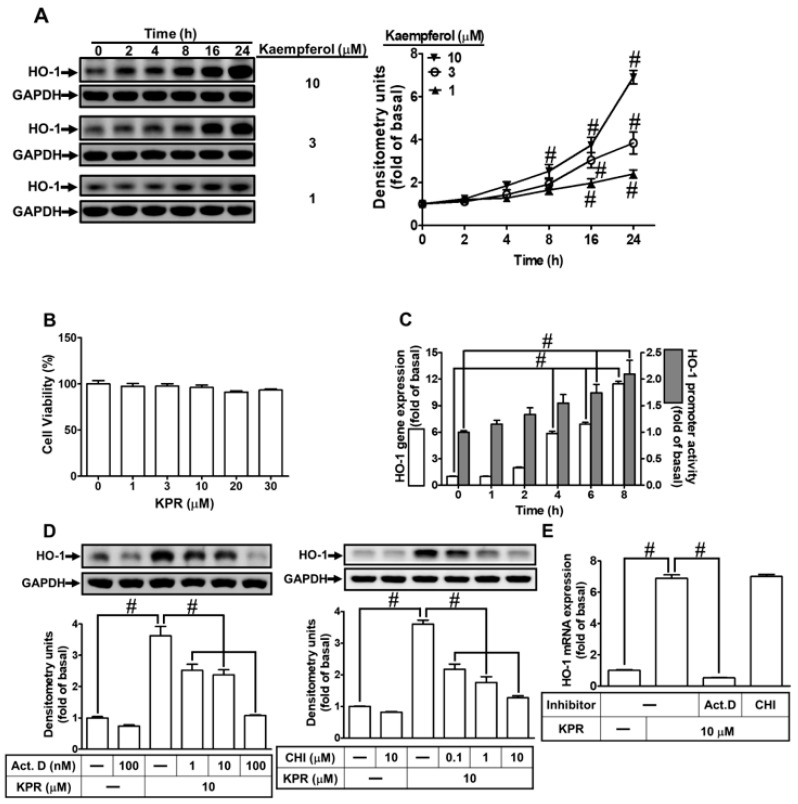
KPR induces HO-1 expression in HPAEpiCs. (**A**) HPAEpiCs were treated with various concentrations of KPR for the indicated time intervals. The protein expression of HO-1 was determined by Western blot using GAPDH as a loading control. (**B**) Cells were treated with various concentrations of KPR for 24 h and the cell viability was examined by an XTT kit. (**C**) HPAEpiCs were treated with KPR (10 μM) for the indicated time intervals. The HO-1 mRNA expression and promoter activity were analyzed by real-time PCR and promoter activity assay kit, respectively. (**D**) HPAEpiCs were preincubated with various concentrations of either Act. D or CHI for 1 h and then incubated with vehicle or KPR (10 μM) for 16 h. The levels of HO-1 protein expression were determined by Western blot using GAPDH as a loading control. (**E**) Cells were pretreated with/without 100 nM Act. D or 10 μM CHI for 1 h, and then incubated with DMSO or KPR (10 μM) for 6 h. The level of HO-1 mRNA expression was analyzed by real-time PCR. Data are expressed as mean ± SEM (*n* = 5), analyzed with one way ANOVA and Dunnett’s post hoc test. # *p* < 0.01, as compared with the cells exposed to vehicle alone; or significantly different as indicated.

**Figure 4 antioxidants-11-00782-f004:**
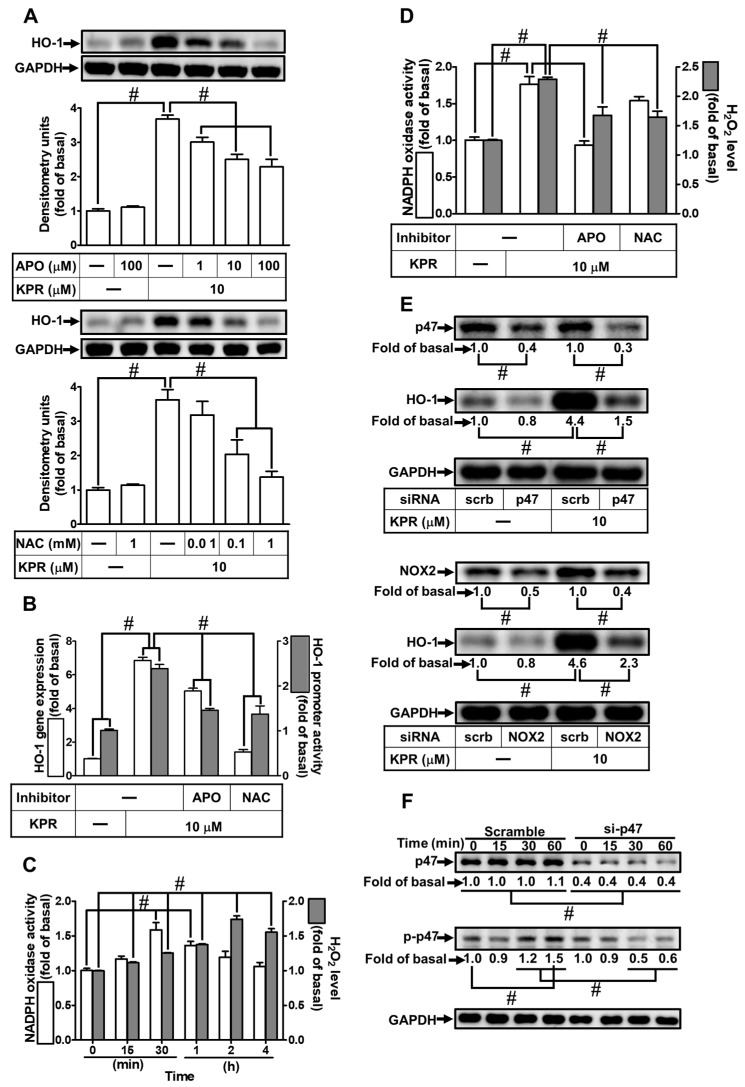
NADPH oxidase activation and ROS generation by KPR regulate HO-1 expression. (**A**) HPAEpiCs were pretreated with various concentrations of APO or NAC for 1 h and then incubated with vehicle or KPR (10 μM) for 16 h. The protein expression of HO-1 was determined by Western blot using GAPDH as a loading control. (**B**) Cells were pretreated with or without APO (100 μΜ) or NAC (1 mM) for 1 h, and then incubated with DMSO or KPR (10 μM) for 6 h or 8 h. The HO-1 mRNA expression and promoter activity were analyzed by real-time PCR (6 h) and promoter activity assay (8 h), respectively. (**C**,**D**) The chemiluminescence was measured for NADPH oxidase activation and ROS accumulation. Cells were treated with KPR (10 μM) at the indicated time intervals. (**D**) Cells were pretreated with APO (100 μM) or NAC (1 mM) for 1 h and then incubated with KPR (10 μM) for the indicated time intervals (30 min for NADPH oxidase activation; 2 h for ROS). (**E**) Cells were transfected with p47 or NOX2 siRNA, and then incubated with KPR (10 μM) for 16 h. The protein levels of HO-1, p47, NOX2, and GAPDH were determined by Western blot. (**F**) Cells were pretreated with p47 siRNA, and then incubated with KPR (10 μM) for the indicated time intervals. The levels of phospho-p47 and total p47 were determined by Western blot. Data are expressed as mean ± SEM (*n* = 5), analyzed with one way ANOVA and Dunnett’s post hoc test. # *p* < 0.01, as compared with the cells exposed to vehicle alone; or significantly different as indicated.

**Figure 5 antioxidants-11-00782-f005:**
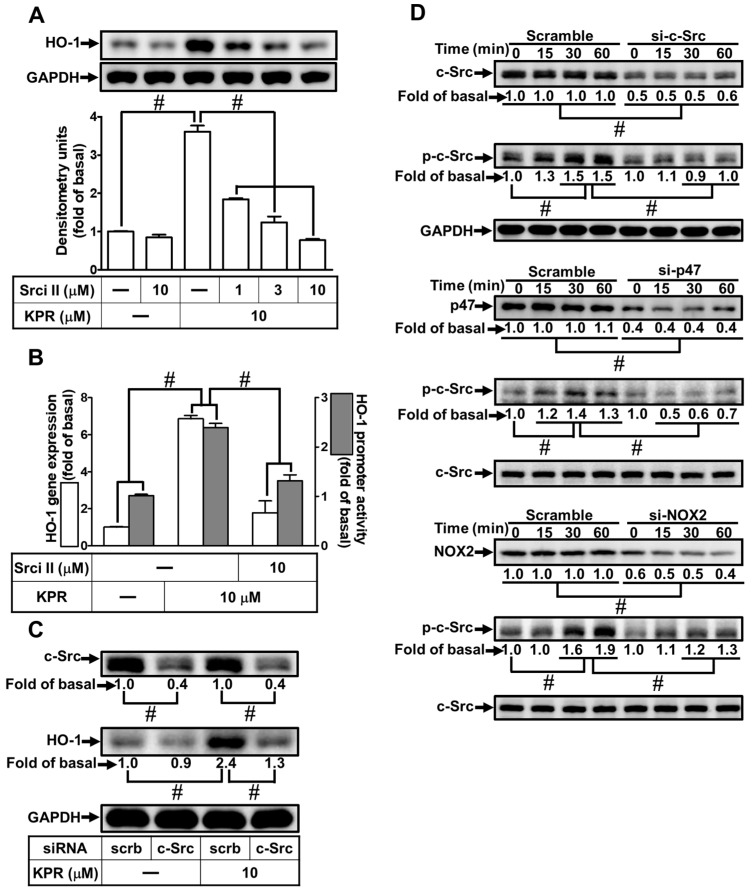
c-Src participates in KPR-induced HO-1 expression. (**A**) HPAEpiCs were pretreated with various concentrations of Srci Ⅱ for 1 h and then incubated with DMSO or KPR (10 μM) for 16 h. The protein expression of HO-1 was determined by Western blot using GAPDH as a loading control. (**B**) Cells were pretreated with or without Srci Ⅱ (10 μM) for 1 h and then incubated with DMSO or KPR (10 μM) for the indicated time intervals. The HO-1 mRNA expression and promoter activity were analyzed by real-time PCR (6 h) and promoter activity assay (8 h), respectively. (**C**) Cells were transfected with scrambled (scrb) or c-Src siRNA, and then incubated with KPR (10 μM) for 16 h. The protein levels of HO-1, c-Src, and GAPDH were determined by Western blot. (**D**) Cells were transfected with c-Src, p47, or NOX2 siRNA, and then incubated with KPR (10 μM) for the indicated time intervals. The levels of phospho-c-Src and total levels of c-Src, p47, and NOX2 were determined by Western blot. Data are expressed as mean ± SEM (*n* = 5), analyzed with one way ANOVA and Dunnett’s post hoc test. # *p* < 0.01, as compared with the cells exposed to vehicle alone; or significantly different as indicated.

**Figure 6 antioxidants-11-00782-f006:**
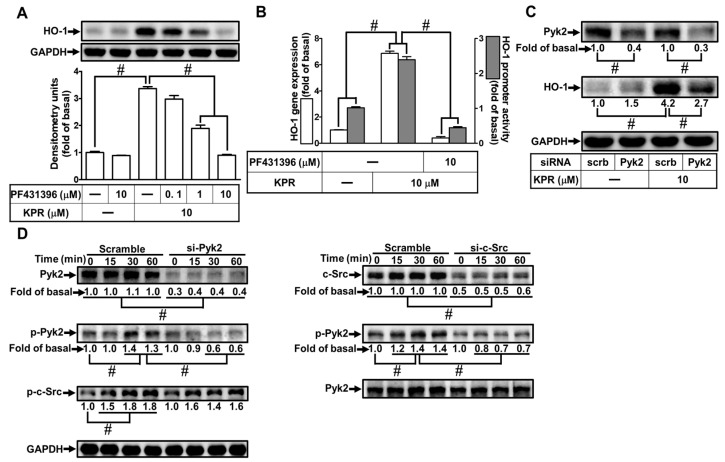
Pyk2 involves in KPR-induced HO-1 expression. (**A**) HPAEpiCs were pretreated with various concentrations of PF431396 for 1 h and then incubated with DMSO or KPR (10 μM) for 16 h. The protein expression of HO-1 was determined by Western blot using GAPDH as a loading control. (**B**) Cells were pretreated with or without PF431396 (10 μM) for 1 h and then incubated with DMSO or KPR (10 μM) for the indicated time intervals. The HO-1 mRNA expression and promoter activity were analyzed by real-time PCR (6 h) and promoter activity assay (8 h), respectively. (**C**) Cells were transfected with scrambled (scrb) or Pyk2 siRNA, and then incubated with KPR (10 μM) for 16 h. The protein level of HO-1, Pyk2, and GAPDH was determined by Western blot. (**D**) Cells were transfected with c-Src or Pyk2 siRNA, and then incubated with KPR (10 μM) for the indicated time intervals. The levels of phospho-Pyk2, total c-Src, and total Pyk2 were determined by Western blot. Data are expressed as mean ± SEM (*n* = 5), analyzed with one way ANOVA and Dunnett’s post hoc test. # *p* < 0.01, as compared with the cells exposed to vehicle alone; or significantly different as indicated.

**Figure 7 antioxidants-11-00782-f007:**
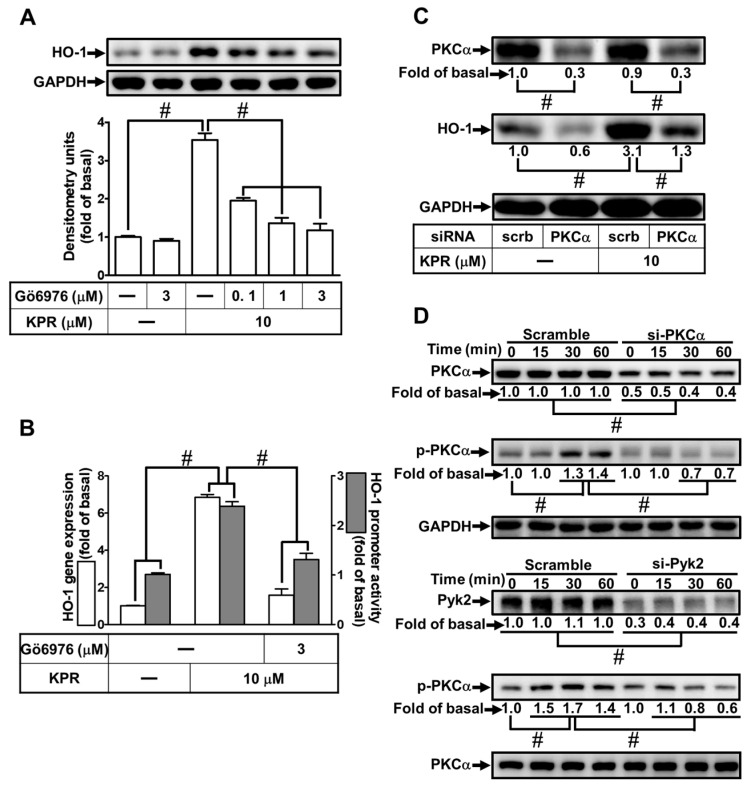
PKCα is required for KPR-induced HO-1 Expression. (**A**) HPAEpiCs were pretreated with various concentrations of Gő6976 for 1 h and then incubated with DMSO or KPR (10 μM) for 16 h. The protein expression of HO-1 was determined by Western blot using GAPDH as a loading control. (**B**) Cells were pretreated with or without Gő6976 (3 μM) for 1 h and then incubated with DMSO or KPR (10 μM) for the indicated time intervals. The HO-1 mRNA expression and promoter activity were analyzed by real-time PCR (6 h) and promoter activity assay (8 h), respectively. (**C**) Cells were transfected with scrambled (scrb) or PKCα siRNA, and then incubated with KPR (10 μM) for 16 h. The protein levels of HO-1, PKCα, and GAPDH were determined by Western blot. (**D**) Cells were transfected with PKCα or Pyk2 siRNA, and then incubated with KPR (10 μM) for the indicated time intervals. The levels of phospho-PKCα, total PKCα, and total Pyk2 were determined by Western blot. Data are expressed as mean ± SEM (*n* = 5), analyzed with one way ANOVA and Dunnett’s post hoc test. # *p* < 0.01, as compared with the cells exposed to vehicle alone; or significantly different as indicated.

**Figure 8 antioxidants-11-00782-f008:**
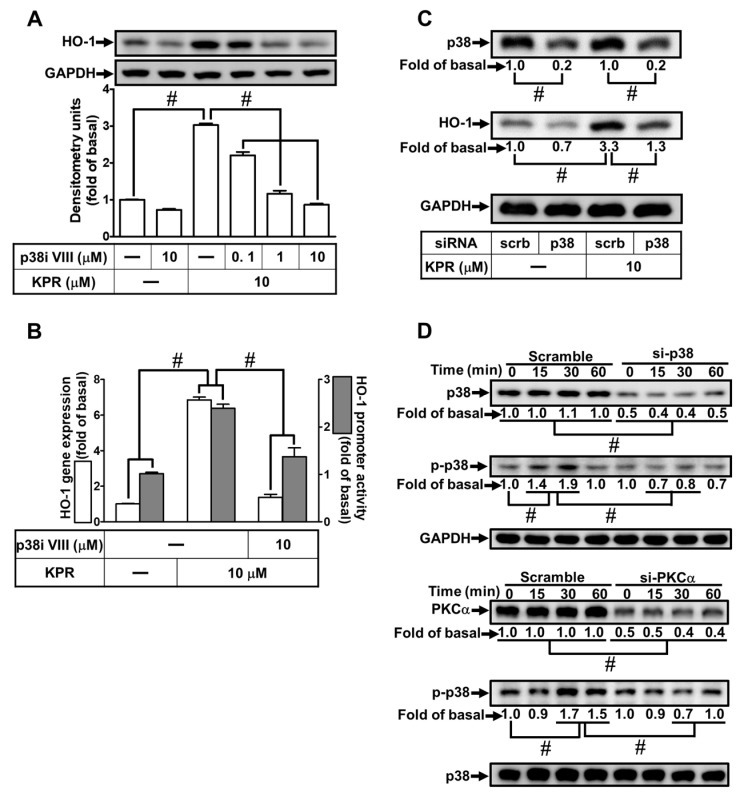
KPR-induced HO-1 expression is mediated through p38 MAKP. (**A**) HPAEpiCs were pre-treated with various concentrations of p38i VIII for 1 h and then incubated with DMSO or KPR (10 μM) for 16 h. The protein expression of HO-1 was determined by Western blot using GAPDH as a loading control. (**B**) Cells were pretreated with or without p38i VIII (10 μM) for 1 h, and then incubated with DMSO or KPR (10 μM) for the indicated time intervals. The HO-1 mRNA expression and promoter activity were analyzed by real-time PCR (6 h) and promoter activity assay (8 h), respectively. (**C**) Cells were transfected with scrambled (scrb) or p38 siRNA, and then incubated with KPR (10 μM) for 16 h. The protein levels of HO-1, p38, and GAPDH were determined by Western blot. (**D**) Cells were transfected with p38 or PKCα siRNA, and then incubated with KPR (10 μM) for the indicated time intervals. The levels of phospho-p38, total p38, and total PKCα were determined by Western blot. Data are expressed as mean ± SEM (*n* = 5), analyzed with one way ANOVA and Dunnett’s post hoc test. # *p* < 0.01, as compared with the cells exposed to vehicle alone; or significantly different as indicated.

**Figure 9 antioxidants-11-00782-f009:**
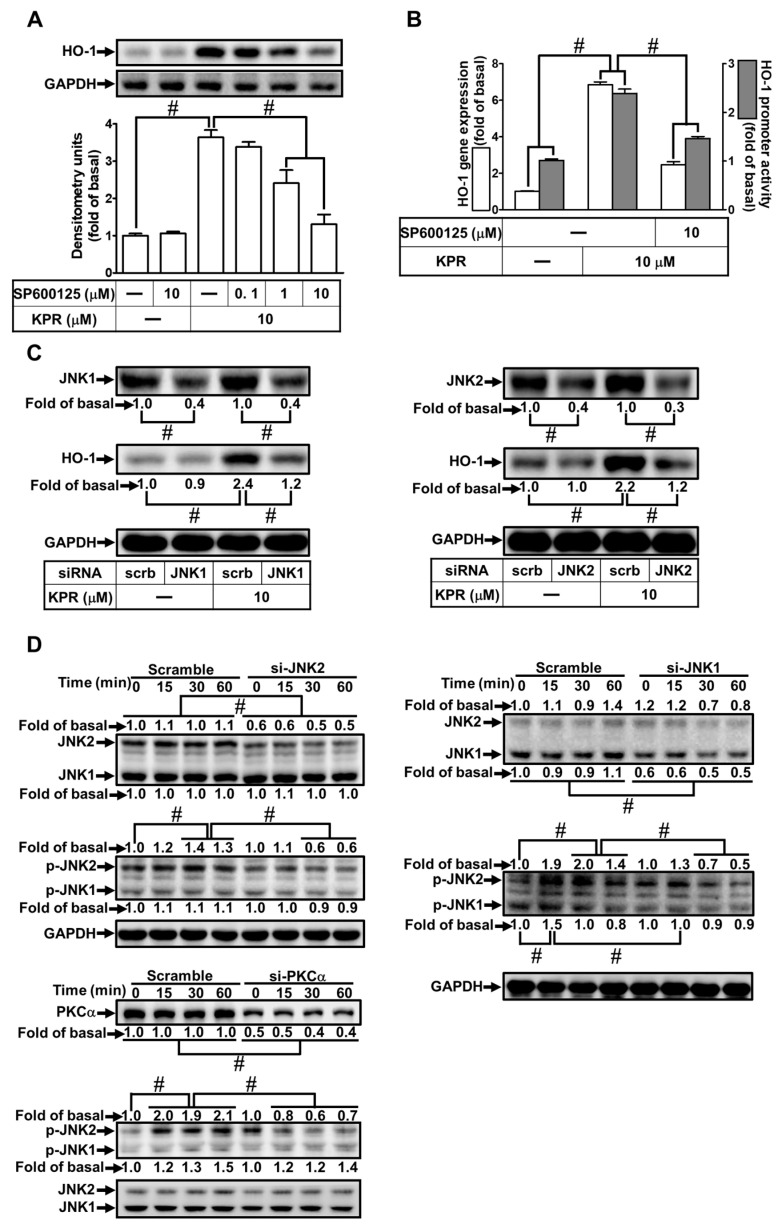
JNK1/2 is necessary for KPR-induced HO-1 expression. (**A**) HPAEpiCs were pretreated with various concentrations of SP600125 for 1 h and then incubated with DMSO or KPR (10 μM) for 16 h. The protein expression of HO-1 was determined by Western blot using GAPDH as a loading control. (**B**) Cells were pretreated with/without SP600125 (10 μM) for 1 h, and then incubated with DMSO or KPR (10 μM) for 6 h or 8 h. The HO-1 mRNA expression and promoter activity were analyzed by real-time PCR (6 h) and promoter activity assay (8 h), respectively. (**C**) Cells were transfected with scrambled (scrb), JNK1, or JNK2 siRNA, and then incubated with KPR (10 μM) for 16 h. The protein levels of HO-1, JNK1/2, and GAPDH were determined by Western blot. (**D**) Cells were transfected with PKCα, JNK1, or JNK2 siRNA, and then incubated with KPR (10 μM) for the indicated time intervals. The levels of phospho-JNK1/2, total JNK1/2, and total PKCα were determined by Western blot. Data are expressed as mean ± SEM (*n* = 5), analyzed with one way ANOVA and Dunnett’s post hoc test. # *p* < 0.01, as compared with the cells exposed to vehicle alone; or significantly different as indicated.

**Figure 10 antioxidants-11-00782-f010:**
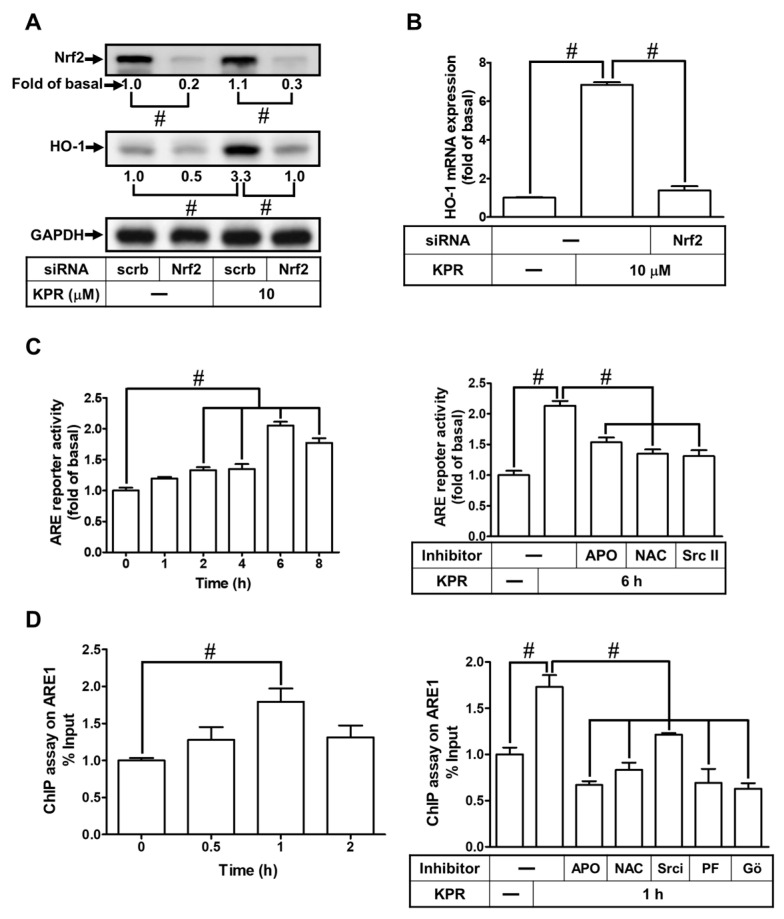
Nrf2 activated by protein kinases is involved in KPR-induced HO-1 expression. (**A**) HPAEpiCs were transfected with scrambled (scrb) or Nrf2 siRNA, and then incubated with DMSO or KPR (10 μM) for 16 h. The protein expression of HO-1 was determined by Western blot analysis using GAPDH as a loading control. (**B**) Cells were transfected with scrambled (scrb) or Nrf2 siRNA, and then incubated with DMSO or KPR (10 μM) for 6 h. The HO-1 mRNA expression was analyzed by real-time PCR. (**C**) HPAEpiCs were co-transfected with ARE promoter-Luc gene and β-galactosidase and then incubated with KPR (10 μM) for the indicated time intervals. The levels of Nrf2 binding with ARE were examined by a luciferase reporter assay kit. Cells were pretreated with APO (100 μM), NAC (1 mM), or Srci II (10 μM) for 1 h and then incubated with KPR (10 μM) for 6 h. ARE promoter activity was determined by a luciferase reporter assay kit. (**D**) Cells were incubated with KPR (10 μM) for the indicated time intervals (upper). The cells were pretreated with APO (100 μM), NAC (1 mM), Srci II (10 μM), PF431396 (10 μM), or Gő6976 (3 μM) for 1 h, and then incubated with KPR (10 μM) for 1 h (bottom). The DNA binding activity of Nrf2 on ARE1 was determined by chromatin immunoprecipitation assay. The immunoprecipitated DNA was analyzed by real-time qPCR with SYBR Green. (**E**) Cells were transfected with scrambled (scrb), p38, JNK2, or Nrf2 siRNA, and then incubated with KPR (10 μM) for the indicated time intervals. The levels of phospho-Nrf2, total Nrf2, p38, and JNK2 were determined by Western blot. (**F**) The cells were pretreated without or with APO (100 μM), NAC (1 mM), Srci II (10 μM), PF431396 (10 μM), Gő6976 (3 μM), SP600125 (10 μM), or p38i VIII (10 μM) for 1 h, and then incubated with KPR (10 μM) for 30 min. The localization and expression of Nrf2 were determined by immunofluorescent staining (green) and nuclei were stained with DAPI (blue). Scale bar: 50 µm. Data are expressed as mean ± SEM (*n* = 5), analyzed with one way ANOVA and Dunnett’s post hoc test. # *p* < 0.01, as compared with the cells exposed to vehicle alone; or significantly different as indicated.

**Figure 11 antioxidants-11-00782-f011:**
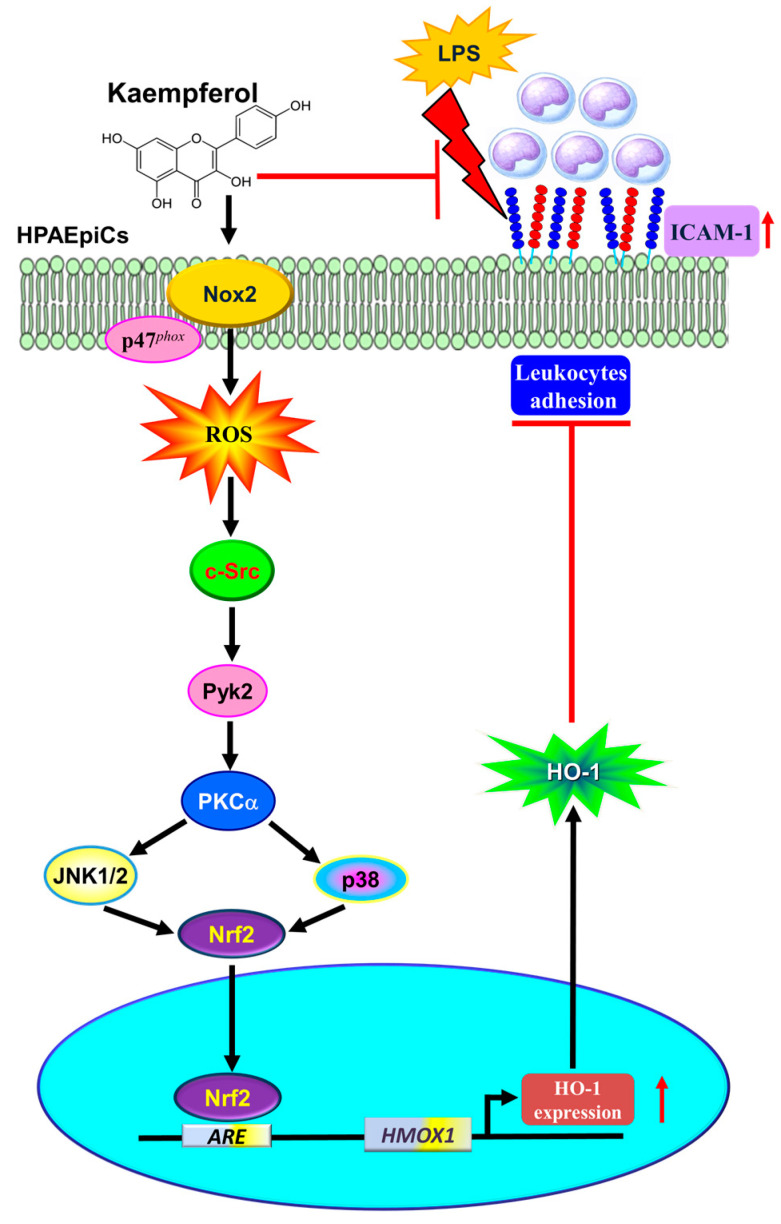
A schematic pathway for KPR-induced HO-1 expression in HPAEpiCs. KPR attenuated LPS-induced ICAM-1 expression and lung monocyte/leukocyte accumulation through upregulation of HO-1 via enhanced p47^phox^/Nox2 activity, resulting in the accumulation of intracellular ROS. Imbalance in oxidative stress promoted the phosphorylation of c-Src/Pyk2/PKCα/p38α MAPK- and JNK1/2-dependent Nrf2 activation, which further binds with ARE on HO-1 promoter and suppresses the LPS-mediated inflammation in HPAEpiCs and in vivo. Thus, upregulation of the HO-1 by KPR exerts a potential strategy to protect against pulmonary inflammation.

## Data Availability

The data used to support the findings of this study are available from the corresponding author upon request.

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
