# Peer review of "HO-1 Upregulation by Kaempferol via ROS-Dependent Nrf2-ARE Cascade Attenuates Lipopolysaccharide-Mediated Intercellular Cell Adhesion Molecule-1 Expression in Human Pulmonary Alveolar Epithelial Cells"

_antioxidants, 2022, doi:10.3390/antiox11040782_

Round 1

Reviewer 1 Report

The manuscript by Yang et al shows that Kaempferol, a flavonoide contained in several pharmaceutically used plants, may be helpful to downregulate ICAM-1 in human alveolar epithelial cells. Such an effect would be interesting as ICAM-1 plays a role in the activation of immune cells including T- and B-cells. Furthermore, ICAM-1 is the major docking protein for rhinovirus, the major cause of common cold, as well as exacerbation in asthma and COPD patients. Thus, the control of ICAM-1 expression by Kaempferol is of interest for the therapy or prevention of lung inflammation.

The manuscript is well written and the concept is sound. However, there are some issues with the second signalling pathway mediated through HO-1.

Major criticisms:

  1. The authors state that Kaempferol downregulates ICAM-1 by upregulating HO-1. The study shows a very detailed analysis of the signalling pathway activated by Kaempferol, but there is not much detail on the regulatory mechanism of HO-1 on ICAM-1. This is even shown in the graphic summary Fig. 11.
  2. In the introduction, the authors describe HO-1 induced signalling in lines 83-94. However, they did not link this to their target ICAM-1. In their experimental settings, there is no data on this signalling pathway, which would be important to confirm that HO-1 is a major regulator of ICAM-1.
  3. Please explain what is shown in Figure 1D. Neither the legend nor the photographs explain what the green is indicating, and how this can be used to measure monocyte adhesion.
  4. Please confirm that the HO-1 Western-blots shown in Figure 4E (WB2 and 5) are not identical with that shown in Figure 5A.
  5. In Figure 10, the contrast needs to be improved since the differences for Nrf2 are not clearly visible.
  6. The abbreviation HPAEpiCs in line 90 has not been introduced in the introduction.
  7. I might suggest to replace references 54-56, all by Lin CC by the more recent publication by the same author in J Clin Med. 2020. Doi: 103390/jcm9010226, which summarised the signalling pathways leading to HO-1 activation.

Author Response

Comments and Suggestions for Authors

The manuscript by Yang et al shows that Kaempferol, a flavonoide contained in several pharmaceutically used plants, may be helpful to downregulate ICAM-1 in human alveolar epithelial cells. Such an effect would be interesting as ICAM-1 plays a role in the activation of immune cells including T- and B-cells. Furthermore, ICAM-1 is the major docking protein for rhinovirus, the major cause of common cold, as well as exacerbation in asthma and COPD patients. Thus, the control of ICAM-1 expression by Kaempferol is of interest for the therapy or prevention of lung inflammation.

The manuscript is well written and the concept is sound. However, there are some issues with the second signalling pathway mediated through HO-1.

Major criticisms:

  1. The authors state that Kaempferol downregulates ICAM-1 by upregulating HO-1. The study shows a very detailed analysis of the signalling pathway activated by Kaempferol, but there is not much detail on the regulatory mechanism of HO-1 on ICAM-1. This is even shown in the graphic summary Fig. 11.

Response: Thank you very much for your invaluable comments and suggestions. In this study, the changes in the levels of ICAM-1 were used as an indicator of functional activity of HO-1 expression by Kaempferol on LPS-induced pulmonary inflammation. We did not explore the detailed mechanisms by which HO-1 inhibits LPS-induced responses. It is an important issue for further study in the future. This is also a limitation of the present study. These statements have been included in the Conclusions.

Please see lines 797-802.

  1. In the introduction, the authors describe HO-1 induced signalling in lines 83-94. However, they did not link this to their target ICAM-1. In their experimental settings, there is no data on this signalling pathway, which would be important to confirm that HO-1 is a major regulator of ICAM-1.

Response: Thank you very much for your invaluable comments and suggestions. Our previous study has made an effort to evaluate the signaling pathway underlying LPS-induced ICAM-1 in human pulmonary alveolar epithelial cells (Cho et al., 2016). As mentioned in Question 1, in this study, the changes in the levels of ICAM-1 were used as an indicator of functional activity of HO-1 expression by Kaempferol on LPS-induced pulmonary inflammation. We did not explore the detailed mechanisms by which HO-1 inhibits LPS-induced responses. It is an important issue for further study in the future. This is also a limitation of the present study. These statements have been included in the Conclusions.

Please see lines 797-802.

Reference cited:

[1] Cho, R.L.; Yang, C.C.; Lee, I.T.; Lin, C.C.; Chi, P.L.; Hsiao, L.D.; Yang, C.M. Lipopolysaccharide induces ICAM-1 expression via a c-Src/NADPH oxidase/ROS-dependent NF-κB pathway in human pulmonary alveolar epithelial cells. American journal of physiology. Lung cellular and molecular physiology 2016, 310, L639-657, doi:10.1152/ajplung.00109.2014.

  1. Please explain what is shown in Figure 1D. Neither the legend nor the photographs explain what the green is indicating, and how this can be used to measure monocyte adhesion.

Response: Thank you very much for your invaluable comments and suggestions. Figure 1D showed that the number of adhesion of THP-1 cells on human pulmonary alveolar epithelial cells challenged with LPS was measured. We have described the meaning of green as an indicator of adhesion of THP-1 cells which were labeled with a fluorescent dye, 10 μM BCECF/AM (detected at 535 nm when the dye is excited at 490 nm). The detailed method of adhesion assay has been described in Method 2.12.

Please see lines, 295, and 300-301, 357.

  1. Please confirm that the HO-1 Western-blots shown in Figure 4E (WB2 and 5) are not identical with that shown in Figure 5A.

Response: Thank you very much for your invaluable comments and suggestions. We supply the original data to clarify these HO-1 Western blots which are different. Please see the original images below.

  1. In Figure 10, the contrast needs to be improved since the differences for Nrf2 are not clearly visible.

Response: Thank you very much for your invaluable comments and suggestions. We have optimized these images in Figure 10F.

Please see Figure 10F.

  1. The abbreviation HPAEpiCs in line 90 has not been introduced in the introduction.

Response: Thank you very much for your invaluable comments and suggestions. We have introduced human pulmonary alveolar epithelial cells (HPAEpiCs).

Please see lines 94-95.

  1. I might suggest to replace references 54-56, all by Lin CC by the more recent publication by the same author in J Clin Med. 2020. Doi: 103390/jcm9010226, which summarised the signalling pathways leading to HO-1 activation.

Response: Thank you very much for your invaluable comments and suggestions. These references (56-58 in the revised version) were cited to explain the role of ERK1/2 in HO-1 induction in various types of cells by a variety of inducers, although ERK1/2 was not involved in HO-1 expression induced by KPR. However, the paper in J Clin Med. 2020. Doi: 103390/jcm9010226 suggested by the reviewer did not evaluate the role of ERK1/2 in HO-1 expression. Thus, we still keep these references in the revised manuscript.

Reviewer 2 Report

With real interest, I read the manuscript antioxidants-1694155. The Authors planned and performed the study very well. The manuscript is nicely written, too, even though it is a bit, it has to be like that with so huge amounts oft he data.

I have only some very minor comments:

C1. Throughout the whole main text manuscript, abbreviations need to be explained upon their first appeaarance (e.g. TNFα, HPAEpiCs).

C2. This should be done separately for the Abstract.

C3. From what I understand, HPAEpiCs are commercially available primary cells. Please, make it clear to the Reader that those are primary cells not a cell line.

C4. In addition, if HPAEpiCs were used, then why THP-1 not primary human monocytes? Please, list as a limitation in the Discussion.

C5. Other study limitations in addition to point 4 above and what you write in lines 723-725?

C6. Lines 315-316. "All the data were expressed as the mean ± SEM, five individual experiments (n 315 = 5).“. Grammar?

C7. The role of airway epithelail cells in respiratory tract disorders such as asthma (PMID: 31904412) and COPD (PMID: 31953012) should be clearly mentioned, the best in the Introduction.

C8. Figure 11 could be used as a graphical abstract as well.

Author Response

Comments and Suggestions for Authors

With real interest, I read the manuscript antioxidants-1694155. The Authors planned and performed the study very well. The manuscript is nicely written, too, even though it is a bit, it has to be like that with so huge amounts of the data.

I have only some very minor comments:

C1. Throughout the whole main text manuscript, abbreviations need to be explained upon their first appearance (e.g. TNFα, HPAEpiCs).

Response: Thank you very much for your invaluable comments and suggestions. We have explained abbreviations upon their first appearances throughout the text.

C2. This should be done separately for the Abstract.

Response: Thank you very much for your invaluable comments and suggestions. We have separately explained abbreviations in the Abstract.

Please see the Abstract on Page 1.

C3. From what I understand, HPAEpiCs are commercially available primary cells. Please, make it clear to the Reader that those are primary cells not a cell line.

Response: Thank you very much for your invaluable comments and suggestions. We have mentioned that HPAEpiCs (primary cells) were isolated from human lung tissues.

Please see line 147.

C4. In addition, if HPAEpiCs were used, then why THP-1 not primary human monocytes? Please, list as a limitation in the Discussion.

Response: Thank you very much for your invaluable comments and suggestions. We used THP-1 cells because of their availability and limited resources for primary monocytes. We understood the differences between these immortalized cell lines THP-1 and their physiological counterparts primary monocytes. We have mentioned this limitation in the Conclusions.

Please see lines 796-797

C5. Other study limitations in addition to point 4 above and what you write in lines 723-725?

Response: Thank you very much for your invaluable comments and suggestions. In this study, the changes in the levels of ICAM-1 were used as an indicator of functional activity of HO-1 expression by Kaempferol on LPS-induced pulmonary inflammation. We did not explore the detailed mechanisms by which HO-1 inhibits LPS-induced responses. It is an important issue for further study in the future. This is also a limitation of the present study. These statements have been included in the Conclusions.

Please see lines 797-802.

C6. Lines 315-316. "All the data were expressed as the mean ± SEM, five individual experiments (n 315 = 5).“. Grammar?

Response: Thank you very much for your invaluable comments and suggestions. We have corrected the sentence as follows: “All the data were expressed as the mean ± SEM, in five individual experiments (n = 5).”

Please see lines 321-322.

C7. The role of airway epithelail cells in respiratory tract disorders such as asthma (PMID: 31904412) and COPD (PMID: 31953012) should be clearly mentioned, the best in the Introduction.

Response: Thank you very much for your invaluable comments and suggestions. We have mentioned the role of airway epithelial cells in respiratory tract disorders as follows:” the airway epithelia are mainly involved in the pathogenesis of these chronic pulmonary disorders [5,6]”.

Please see lines 50-51.

References cited:

  1. Hadzic, S.; Wu, C.Y.; Avdeev, S.; Weissmann, N.; Schermuly, R.T.; Kosanovic, D. Lung epithelium damage in COPD - An unstoppable pathological event? Cellular signalling 2020, 68, 109540, doi:10.1016/j.cellsig.2020.109540.
  2. Potaczek, D.P.; Miethe, S.; Schindler, V.; Alhamdan, F.; Garn, H. Role of airway epithelial cells in the development of different asthma phenotypes. Cellular signalling 2020, 69, 109523, doi:10.1016/j.cellsig.2019.109523.

C8. Figure 11 could be used as a graphical abstract as well.

Response: Thank you very much for your invaluable comments and suggestions. We have prepared a graphical abstract based on the concept of Figure 11.

Please see the Graphical Abstract.

Round 2

Reviewer 1 Report

no further actions needed.